## REVIEW ARTICLE

# From whole-mount to single-cell spatial assessment of gene expression in 3D

Lisa N. Waylen[1], Hieu T. Nim[1,2], Luciano G. Martelotto[3] &
Mirana Ramialison 📧[1,2]✉

Unravelling spatio-temporal patterns of gene expression is crucial to understanding core biological principles from embryogenesis to disease. Here we review emerging technologies, providing automated, high-throughput, spatially resolved quantitative gene expression data. Novel techniques expand on current benchmark protocols, expediting their incorporation into ongoing research. These approaches digitally reconstruct patterns of embryonic expression in three dimensions, and have successfully identified novel domains of expression, cell types, and tissue features. Such technologies pave the way for unbiased and exhaustive recapitulation of gene expression levels in spatial and quantitative terms, promoting understanding of the molecular origin of developmental defects, and improving medical diagnostics.

Spatial control of gene expression is crucial for defining tissue identity, from the patterning of a zebra's stripes, to the left-right asymmetry present in most species, and deviations from this regulatory programme may result in developmental defects or disease, such as situs inversus or the variation in antigens expressed at different layers of a cancer tumour.

Our body plan relies on spatial expression, achieved by correct deployment of a developmental gene regulatory network (GRN) where the location, timing, and level of developmental gene expression are crucial. For instance, Hox genes are a highly conserved family involved in body axis patterning and provide an illustration of the requirement for specific timing, spatial restriction, and expression degree across species. Hox genes are silenced until activated by developmental timing cues, and sequential expression of Hox clusters will define the body plan of the embryo[1–3]. Ectopic expression of *Hoxa-1* in mice can reorganise regions of the hindbrain by transforming developing rhombomeres[4], and in zebrafish, misexpression of *Gata5* induces ectopic expression of myocardial genes such as *nkx2.5*[5]. Of particular importance in development are morphogens, a concentration gradient dependent signal that provides a pattern for tissue differentiation and development. While some genes employ a simple on/off mechanism, many developmental genes respond to different expression thresholds to provide phenotypically distinct outcomes. In *Drosophila*, the maternal gene *bicoid* is responsible for regulating

---

[1] Australian Regenerative Medicine Institute and Systems Biology Institute, Monash University, Clayton, VIC, Australia. [2] Transcriptomics and Bioinformatics Group, Murdoch Children's Research Institute, Parkville, VIC, Australia. [3] Single Cell Core Laboratory, Harvard Medical School, Department of System Biology, Boston, MA, USA. ✉email: mirana.ramialison@monash.edu

development of the anterior pole through graduated mRNA diffusion[6] and limb patterning in the chick is established by morphogenic fields[7].

Organoids raised from human stem cell cultures provide an invaluable model for exploring disease mechanisms spatially, by uncovering the cellular and molecular interactions occurring in three dimensions (3D). For instance, embryonic models such as gastruloids[8], and subdomains of organs such as the kidney can be recapitulated from the preferential induction of progenitors[9]. This offers a unique platform to capture the spatial gene expression information required for accurate tissue patterning. Deviations from programmed, spatio-temporal gene expression can lead to disease, as seen in congenital disorders[10] and highly heterogeneous cancer tumours which provide a prime target for spatial profiling analysis. For example, the dysregulation of HOX genes in leukaemia has been shown to support the immortalisation of malignant cells, while specific spatial expression patterns of biomarkers inform clinical prognosis and therapies[11,12]. In plants, changes in spatial gene expression have been shown to provide advance warning for the progression of viral infection[13].

Therefore, to accurately decipher the GRNs involved in spatial gene expression, it is necessary to capture gene identity along with quantitative data. Previously, the Allen Mouse Brain Atlas constructed by the Allen Institute for Brain Sciences[14] integrated bulk and computational methods with in situ hybridisation gene expression data, later accompanied by human and macaque atlases[15,16]. Despite challenging low resolution and lack of quantitative data, these atlases significantly advanced the current understanding of brain structure and function. More recently, the Allen Mouse Brain Common Coordinate Framework (CCFv3)[17] sampled 1675 mouse brains at 10 μm voxel resolution, integrating multiple datasets in 3D. Virtual atlases are publicly available, browsable cell reference maps where gene expression data is reconstructed in 3D, digital space. They support myriad fields of research through interactive visualisation and analysis, providing a valuable source of information for many model organisms including *C. elegans* (Wormbase)[18], frogs (Xenbase)[19], *Drosophila*[20] and zebrafish[21].

This review explores current techniques used to elucidate spatial expression, from established robust methods with throughput limitations, to cutting-edge systematic and unbiased measurement of gene expression (Fig. 1, Table 1). We explain the power of accurately capturing spatial gene expression data quantitatively, to decipher molecular interactions driving major biological processes.

## Imaging-based methods for resolving gene expression spatially

Historically, spatial patterns were captured from direct or in situ staining of genes of interest. In situ hybridisation (ISH) techniques (Fig. 1a) were developed to study spatial complexity. This involves hybridisation of a labelled RNA to an endogenous mRNA transcript within the cell[22]. Large-scale in situ hybridisation is routinely used to simultaneously interrogate expression patterns of genes in sections or whole organs and tissues[23,24]. This is a particularly efficient method to screen for genes with regionalised expression patterns, such as those responsible for early growth determination[25], but these techniques are low-throughput. At most a hundred genes can be simultaneously assessed, albeit in different samples. Another challenge of this traditional model is that visualisation of expression relies on probe binding efficiency, which varies significantly, eliminating the ability to perform comparative study between tissues or genes, without first performing background neutralising or equalising equations[26]. While in situ hybridisation addresses the need for spatially resolved data, accurate quantification of gene expression is often difficult to achieve using this approach and therefore requires further analysis.

**Increasing sensitivity and accuracy**. Further advancements to accurate quantitative measurement of in situ gene expression include the Spatial Genomic Analysis pipeline (SGA)[27], which integrates the output from sequential single-molecule fluorescence in situ hybridisation and hybridisation chain reaction (HCR). HCR employs two complementary DNA probes which form a fluorescently labelled amplification polymer capable of relative quantification of mRNA expression[28]. In addition, this technology achieves higher spatial resolution due to the integrated single-cell imaging techniques. The SGA pipeline allowed novel characterisation of a pluripotent stem cell niche in development of the dorsal neural tube[27]. *RNAscope* provides a further integration of qualitative and quantitative approaches whereby probe hybridisation and signal amplification are performed in situ, allowing spatially relevant analysis at single-molecule resolution[29]. However, analysis is limited to a handful of target genes, since throughput from SGA is restricted by the number of distinguishable dyes.

Click-amplifying fluorescent ISH *(ClampFISH)* was developed to improve the efficiency in FISH and the resulting weak signal response that demands high power microscopy. Indeed, the enzymes required in FISH may prove inefficient, providing inconsistent amplification or poor diffusion through the cell. *ClampFISH* detects unique nucleic acid molecules, and allows padlock-style probes to be used without enzyme support. Click chemistry is used to covalently circularise amplifier molecules by linking 5′ and 3′ ends[30], topologically entangling individual amplifier molecules to increase binding specificity.

Recent advances in ISH have introduced a novel approach to improve signal detection. Single-molecule fluorescence in situ hybridisation (*smFISH*) relies on the use of many short probes which target multiple regions of the mRNA transcript. This can approximate the level of gene expression between different tissues, but is not numerically quantitative[31]. It is also limited by the need to bind sufficient primers to the query mRNA, which may not be possible for many short mRNAs. Although *smFISH* cannot assess non-coding RNAs such as those required for many regulatory functions, as it targets the ribosome/mRNA interactions, it can provide spatial localisation of RNA expression by directly imaging individual RNA molecules in single cells. Throughput is limited by the number of RNA species that can be simultaneously measured in single cells.

While previous studies in *smFISH* have improved the number of probed genes accessible via a targeted, barcode approach, a new cyclic technique, ouroboros *smFISH* (*osmFISH*) relies on a non-barcoded approach focussed on designing image processing tools capable of processing large tissue areas and extensive data sets[32]. *smFISH* is more efficient at detecting RNA than single-cell RNA sequencing (scRNA-seq)[33], and *osmFISH* has combined short hybridisation times and background reduction to attain high signal-to-noise ratios to further improve the sensitivity of RNA detection. *osmFISH* achieved a rate of 30% zero counts in contrast to the 82% registered by scRNA-seq, indicating its higher sensitivity to recover low levels of gene expression. This is largely because target number scales with the number of required hybridisation rounds to reduce error, and highly expressed genes do not affect the detection of non-highly expressed genes as barcoding is not required, and images can be analysed separately. FISH approaches rely on downstream image analysis to extract numerical measurements of gene expression. Meanwhile, scRNA-seq also requires reads filtering, mapping, and quality control to

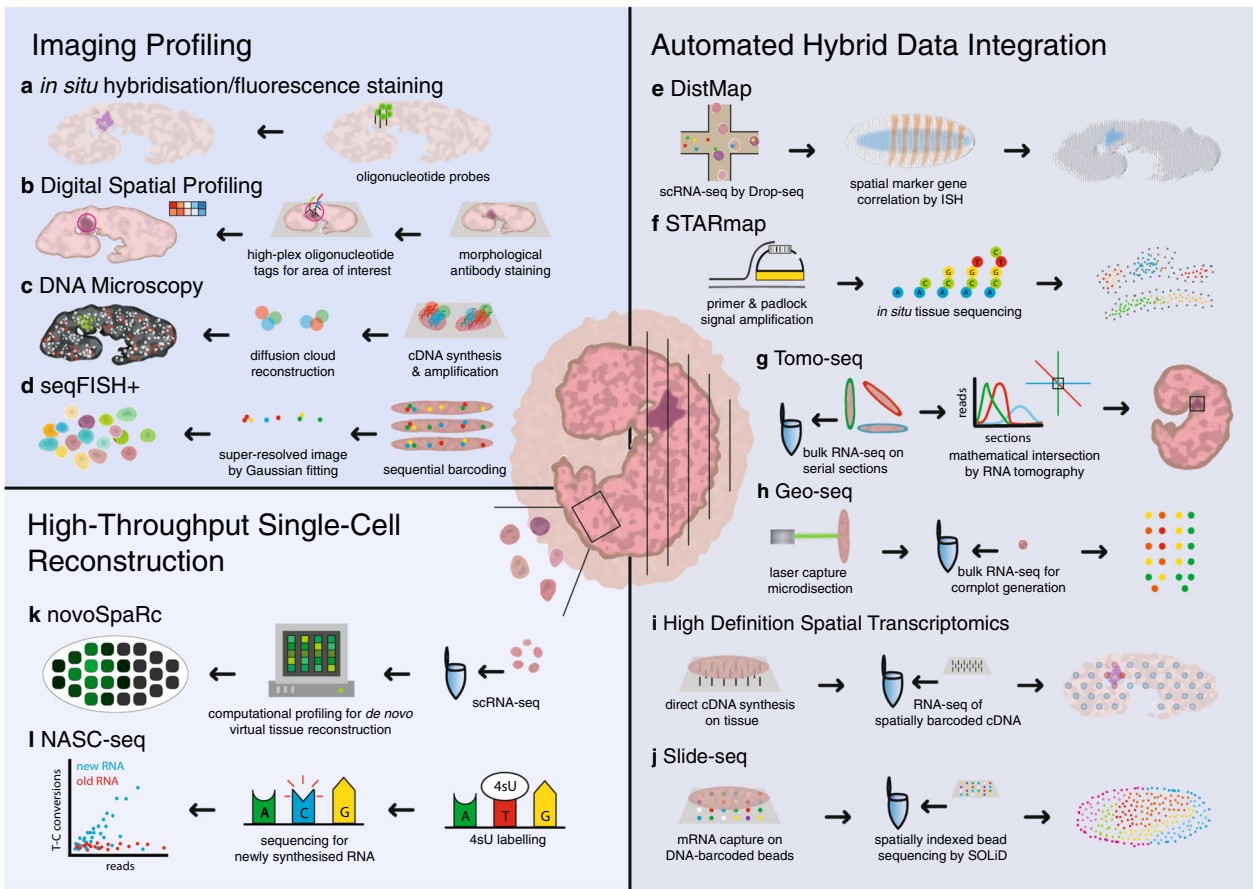

**Fig. 1 Principles of current methods for capturing spatial gene expression.** Schematic overview of methods based on imaging profiling of the entire specimen. **a** In situ hybridisation/fluorescence staining where bound oligonucleotide probes reveal spatial expression via fluorescent dyes. **b** Digital Spatial Profiling where digital barcodes tag bound oligonucleotides and allow multiplexed spatial profiling. **c** DNA microscopy where chemical DNA reactions permit spatial imaging. **d** seqFISH+ where accurate fluorescent barcoding is performed sequentially to improve throughput and generate spatial atlases in situ. **e** DistMap where Drop-seq technology integrates scRNA-seq data and ISH imaging to reveal spatial gene expression. **f** STARmap where genes are sequenced in situ using padlock amplification. **g** Tomo-seq where cryogenic tissue sections are individually analysed by bulk RNA-seq and spatial data triangulated in three axes. **h** Geo-seq where cryogenic tissue samples are obtained through laser capture microdissection and analysed through bulk RNA-seq with results spatially mapped. **i** High-definition spatial transcriptomics where cDNA synthesis is performed in situ and spatially barcoded prior to RNA-seq. **j** Slide-seq where mRNA is barcoded in situ and spatially indexed by SOLiD. **k** novoSpaRc where scRNA-seq is digitally profiled to virtually reconstruct the tissue. **l** NASC-seq where 4sU labelling identifies temporal and spatial features of single-cell data. Single-cell RNA-seq (scRNA-seq), in situ hybridisation (ISH), sequencing by oligonucleotide ligation and detection (SOLiD).

be converted into numerical measurements. *osmFISH* has been optimised for thin tissue sections, and a study of murine brain tissue developed an atlas of gene expression in the somatosensory cortex using 33 marker genes. Although a majority (59%) of the data came from neurons, many different cell types were observed and characterised.

**Increasing throughput and resolution.** Multiplexed error-robust fluorescence in situ hybridisation (MERFISH) multiplexes *smFISH*[34] and assigns barcodes to individual RNA species, which can be imaged sequentially, allowing single-cell transcriptomic profiling. Branched DNA (bDNA) amplification substantially improved MERFISH detection efficiency, without increasing fluorescent spot size for individual transcripts, and it is now possible to use this method to study short RNAs[35]. This approach generates in situ transcriptomic analyses and detects >1000 RNA species, and provides error correction for >100. Further improvements to the efficiency of MERFISH gene processing allow simultaneous imaging of ~10,000 genes at a detection efficiency of ~80%[36]. While providing a high standard of

subcellular resolution, the necessity for cell-dissociation challenges the spatial power of this approach.

Digital Spatial Profiling (Fig. 1b), a new platform from Nanostring, allows multiplexed analysis of protein or RNA (~100 and ~1000plex, respectively), capable of single-cell resolution. Probes coupled with photocleavable, spatially barcoded tags can be used to simultaneously profile targets from a sample tissue region. Early testing suggests that this will be a powerful new tool in understanding the tumour microenvironment and identifying unique, prognostic patient biomarkers[37]. Reproducibility between sections is high, and result validation demonstrates a fair correlation with IHC, FCM and QIF, with some spatially restricted expression patterns of immune markers found to be associated with patient outcome[38]. A key advantage of this approach is that it is non-destructive, permitting reanalysis, and analysis of protein and RNA in cohort on serial sample sections. Digital Spatial Profiling is currently limited, with only four fluorescent visualisation stains available to examine tissue morphology and determine regions of interest.

Traditional spatial mapping relies on fluorescence light microscopy or direct physical registration of transcripts. DNA

**Table 1 Features of current methods for capturing spatial gene expression.**

| Methodology | Coverage | Number of genes | Number of cells[a] | Spatial resolution |
|---|---|---|---|---|
| In situ hybridisation[22] | Targeted | 3 | Low | Tissue |
| RNAscope[29] | Targeted | 12 | Low | Cellular |
| ClampFish[30] | Targeted | 3 | Low | Subcellular |
| smFISH[33] | Targeted | 3 | Low | Subcellular |
| osmFISH[32] | Targeted | 1–33+ | Low | Subcellular |
| MERFISH[35] | Targeted | 10,000 | Medium | Subcellular |
| DNA microscopy[39] | Targeted | | Low | Cellular |
| seqFISH+[40] | Targeted | 10,000 | High | Subcellular |
| DistMap[50] | Targeted | 8000+ | High | Cellular |
| STARMap[51] | Targeted | 1020+ | High | Cellular |
| Tomo-seq[52] | Transcriptome-wide | Whole transcriptome | High | Cellular |
| Geo-seq[56] | Transcriptome-wide | Whole transcriptome | High | Cellular |
| Spatial transcriptomics/10X Visium[59] | Transcriptome-wide | Whole transcriptome | Medium | 100 μm/55 μm |
| Slide-seq[61] | Transcriptome-wide | Whole transcriptome | Medium | 10 μm |
| HDST[63] | Transcriptome-wide | Whole transcriptome | High | 2 μm |
| novoSpaRc[69] | Transcriptome-wide | Whole transcriptome | High | Cellular |
| NASC-seq[72] | Transcriptome-wide | Whole transcriptome | High | Cellular |

[a]Number of cells—low: 0–100, medium: 100–1000, high: 1000–10,000+.

microscopy (Fig. 1c) instead, involves chemical reactions whereby transcript molecules are tagged in situ with random nucleotides, labelling each uniquely[39]. A second reaction amplifies these tagged molecules, concatenates the copies, and adds new random nucleotides. While high signal intensity is a drawback to fluorescent imaging techniques, increased signal density is beneficial to DNA microscopy. Molecular proximities are computationally decoded from the overlapping diffusion fields generated by these signals, from which it is possible to infer physical images of the original transcripts at cellular resolution. Sequencing power permits detection of single-nucleotide variation, and spatially resolves biological features such as somatic mutation and stochastic RNA splicing. Variation in molecular density, however, makes it difficult to reconstruct images spanning large distances. Early applications indicate that DNA microscopy can function as an imaging medium equivalent to optical analysis, and is an important emerging approach to study the complex chemical dynamics of biological systems.

FISH-based approaches allow transcripts to be directly labelled within native tissue sections. In regions of dense transcript concentration, molecular crowding leads to the spatial overlap of fluorescence signals during simultaneous transcript imaging. Extended sequential FISH (seqFISH+) serially reprobes a single slide to generate multiple images which can be computationally merged and analysed for a complete library of transcripts (Fig. 1d), reducing the impact of optical crowding as only 1/60th of the transcript is visualised in each image[40]. In one study, 10,000 genes were analysed in cultured mouse cells. The subsequent gene expression profiles were quantitatively equivalent to RNA-seq data. Within the brain, cells were characterised based on their expression profile, revealing the expected tissue layers when analysing the spatial relationships. Ligand-receptor encoding RNA pairs observed in adjacent cells indicated functional relationships which were dependent on local tissue context/positioning. Comparison with the 60 genes analysed from smFISH showed that detection efficiency of seqFISH+ was 49% which is highly sensitive compared with scRNA-seq.

In summary, in situ mapping of spatial gene expression captures both RNA quantity and position. It has overcome challenges to maintain efficiency, signal intensity and accuracy while scaling to large gene numbers. However, these approaches are still limited in their throughput and ability to accurately measure the level of gene expression.

## Next-generation sequencing-based methods for resolving gene expression spatially

Quantitative methods of measuring accurate gene expression levels in high-throughput have been developed such as CAGE, SAGE, and gene expression arrays. CAGE sequencing permits analysis of coding and non-coding RNA expression by targeting promoter regions instead of translation interactions[41]. Massive parallel RNA sequencing (RNA-seq) allows a high-throughput accurate measurement of RNA transcript levels that can be achieved in an organ (bulk)[42] or single cells through read-counts[43]. Data from bulk tissue analysis may mask the expression patterns of rare, or equally dominant cell types, or omit transient expression patterns. The quantitative data gained in RNA-seq is contrasted by the loss of spatial information. When isolated cells are spatially dislocated from their tissue of origin, transcripts lose spatial context; however, some studies have attempted to address these issues by using downstream cell sorting which reunites distinct cell populations. To achieve an analysis which benefits from data obtained from both spatial and quantitative methods, the two data sets need to be combined, a process which remains largely manual. Recent experiments to combine spatial information, with the quantitative data obtained from RNA-seq have been used to query the transcriptional changes in different regions of the cardiac infarct zone following heart injury in adults and neonates, providing type-specific quantitative expression data[44].

**Automated hybrid approaches**. Mathematical models allow the integration of spatial and quantitative results by associating 'landmark' reference genes with transcript counts. For instance, Seurat is an algorithm that infers cellular localisation by integrating scRNA-seq data with in situ expression patterns[45]. It relies on spatial segregation of landmark genes, identified through ISH experiments, to construct a reference map by including the RNA-seq data to quantify the expression level of these landmark genes. This map provides a transcriptome-wide reflection of spatial patterning which was used to identify key cells subpopulations in the *Drosophila* eye, including interommatidial cells and photoreceptors, each exhibiting a unique set of differentially expressed genes[46].

Moreover, one of the primary problems associated with scRNA-seq analysis, is the high technical noise of small-mass embryos, and subsequent low-copy number transcripts. To

overcome this, the computations in Seurat also consider the spatial information of multiple genes that are co-regulated with landmark genes, thus providing a reliable gene expression map. Another computational method which integrates ISH with scRNA-seq provides a high-throughput approach to identify the tissue of origin for complex tissues[47]. A selection of brain cells from marine annelid *P. dumerilii* was profiled, and 81% of these cells were able to be mapped back to precise spatial locations. This approach relies on knowledge of ISH gene expression, and may not thoroughly encompass all embryonic domains. Both computational methods are powerful tools to generate data that is both spatially resolved, and quantitatively significant. However, these results are algorithm-based, rather than direct reflections of in situ measurements of the embryo.

*Drosophila melanogaster* is, traditionally, a strong model for patterned expression with early emergence of a spatially defined embryo. Using a technique employing low-input RNA-seq, cryosectioned *Drosophila* embryos along the AP axis and isolated and sequenced mRNA, spatial patterns were observed to closely match previously determined in situ and microscopy patterns[48]. This method was used to generate a time course of the developing embryo. Many genes not previously characterised as spatial genes were found to exhibit spatially restricted expression, such as pole specific genes with no functional annotation. This approach experienced a large degree of noise, as too much carrier RNA gave only a small number of reads per slice. Expression was measured across spatially resolved tissue slices, leading to the blending of cell signals and providing poor resolution of cell populations that did not reach the single-cell level. While spatial landmarks could be identified, spatially intermixed populations could not be well defined. Computational approaches can be employed to spatially reconstruct complex tissues without needing prior annotation, a process used to investigate maternal factors with mutants[49], however, this still provided lower resolution results than in situ analysis, and it was difficult to observe transitional domains.

In order to encapsulate the complexity of developmental GRNs in a digitally reconstructed embryo model, it is necessary to perform genome-wide transcriptomics at single-cell resolution. A recent approach analysed 84 well-characterised in situ gene markers in combination with high-throughput, quantitative Drop-Seq to generate computational based algorithm *DistMap* (Fig. 1e), able to confidently resolve 87% of cells in the stage 6 *Drosophila* embryo[50]. This revealed novel roles for several transcription and signalling factors, which had previously not been implicated in early development, and demonstrated the importance of non-coding RNAs in these pathways. Most cells analysed exhibited unique transcriptomes, highlighting the necessity for single-cell resolution.

A key benefit of scRNA-seq is the power to detect biological variability between cells, and define characteristics of rare cell types. While in situ RNA-seq is able to provide spatial context, throughput efficiency is limited and is difficult to scale to whole tissues. The *STARmap* (Fig. 1f) approach promises to deliver gene expression in 3D to cellular resolution. Recent in situ hybridisation methods enable high-resolution imaging of RNA transcripts in intact tissue by exploiting hydrogel-tissue chemistry (HTC) to link in situ synthesised polymers[51]. All cellular RNAs are labelled with two probes, one with a five-base barcode, providing a gene-unique identifier for later multiplexed gene detection. Both probes need to hybridise to the same RNA molecule to reduce noise, and for enzymatic amplification which generates a DNA nanoball with multiple copies of cDNA probes. This process decodes the DNA sequence into multi-coloured fluorescence signals ready to be imaged. Using this two-base sequencing system, >1000 genes are sequenced over 6 imaging cycles. In addition, sequencing with error-reduction by dynamic annealing and ligation (SEDAL) is effective in rejecting errors. Using this approach, a 160-gene set was detected and quantified in mouse primary visual cortex, where clustering revealed distinct cell types, overlaid with spatial expression distribution across layers of cortex. Upregulation of activity-regulated genes in response to visual stimulation was observed. Gene expression results were found to correlate well to in situ hybridisation and scRNA-seq. This technique is scalable to larger 3D tissue blocks and can be adapted for higher gene numbers, with current best throughput at ~1000 genes. The challenge to sequence all genes simultaneously remains.

**High-throughput tissue sectioning-based approaches**. *Tomo-seq* (Fig. 1g) automatically generates spatial resolution equivalent to an in situ hybridisation with the quantitative strength of RNA-seq[52]. The tissue of interest is cryosectioned in three different orientations (antero-posterior, dorso-ventral and lateral planes), followed by bulk RNA-seq of each section. Triangulation algorithm, RNA-tomography, is then automatically applied to estimate gene expression levels at the intersection of the three sections, creating a digital three-dimensional expression pattern of the whole embryo. Depending on section thickness, this mathematical reconstruction could recapitulate gene expression at a resolution equivalent to single-cell analysis. However, the accuracy of this computed gene expression value needs to be validated with scRNA-seq datasets. *Tomo-seq* necessitates using identical samples for an accurate overlay of sections on multiple axes. As this is biologically difficult to accomplish, there is room for a degree of error in combining tissue sections. Applied to developing zebrafish embryos at three developmental stages (shield, 10 somites and 15 somites stages), *Tomo-seq* delivers a comprehensive digital resource for embryonic gene expression in the whole embryo. Approximately 20% of the zebrafish transcriptome exhibited spatially restricted gene expression, a large number of which were functionally uncharacterised[52]. Combined with chromosome conformation capture datasets, *Tomo-seq* was used to identify spatially co-expressed genes correlating with topological domains[53]. Pre-zygotic transcription exhibited strict structuring with super enhancers clustering during development, a pattern which is similar to that previously observed in mammals. Super enhancer regions present a greater degree of transcriptional regulation, and are associated with more highly expressed genes[54]. *Tomo-seq* has also been successfully applied to an entire isolated organ. Indeed, a high spatial resolution map of the embryonic zebrafish heart obtained with *Tomo-seq* revealed 1100 genes being differentially expressed in the various sub-compartments of the developing heart, 502 of which were previously uncharacterised. Further examination of the sinoatrial pacemaker region of the heart revealed a novel role for Wnt/β-catenin signalling in the pathway regulating autonomic control, which provided useful insights into the molecular basis of heart function. This illustrates the discovery potential of this technique to search for novel gene targets for therapeutic use[55].

A complementary method, *Geo-seq* (Fig. 1h), cryosections organs or tissues of interest in one direction only. Each section is sampled by laser capture microdissection, generating cell clusters (~10 cells per cluster), which are then subjected to bulk RNA-seq[56]. As the spatial origin of each cell cluster is uniquely identified, this allows reconstruction of spatial distribution of gene expression within the whole tissue using the Zipcode Mapping Feature. *Geo-seq* was applied to gastrulating mouse embryos to generate a virtual whole-mount 3D model of the spatial expression of all 20,000 mouse genes at this stage[57]. This iTranscriptome was central in revealing novel molecular markers for different cell lineages, for instance, highlighting the role of Oct4 in cell fate determination and A–P axis patterning[58].

*Spatial transcriptomics* (Fig. 1i) is the least biased method to capture spatial gene expression from sections. cDNA is directly synthesised from fixed tissue sections on 6.2 mm × 6.6 mm pucks and is concomitantly labelled with molecular barcodes which record the spatial location of the transcript within the section[59]. The cDNA is then sequenced by RNA-seq. This approach eliminates the need for cell isolation, and was able to detect low-level transcript expression and capture more genes and transcripts than laser capture microdissection, and with higher sensitivity and resolution compared to *Tomo-seq* and *Geo-seq*. The ability to generate histological data which precisely complements the tissue section involved in quantitative analysis is unique to *Spatial transcriptomics* (microarray-based technology) and not possible with NGS-based methods. By processing different tissue domains within the same reaction, *Spatial transcriptomics* is able to remove much of the technical variation experienced by many standard expression methods. Application of this technique in murine brain tissue demonstrated gene clustering that correlated with morphological layers, allowing identification of cluster (and subsequently tissue) specific markers, while analysis of human breast cancer cells revealed that invasive cells exhibited high expression of extracellular-associated genes, and an unexpectedly high degree of cellular heterogeneity within biopsy samples, information important for decisions about prognosis[59]. The integration of *Spatial transcriptomics* with scRNA-seq from dissociated whole human embryonic hearts was able to identify a novel population of Myoz2-enriched cardiomyocytes in both the atria and ventricles which complemented recent findings in mice, yet to be observed in humans[60]. Currently, the resolved power of *Spatial transcriptomics* remains limited by the spacing of primers required for barcoding.

*Slide-seq* (Fig. 1j) alleviates this limited resolution using DNA-barcoding beads bound to 3-mm slides and exposed to fresh tissue sections releasing mRNA from which barcode sequences can be determined with sequencing by oligonucleotide ligation and detection (SOLiD)[61]. This variant of Drop-seq[62] allowed the identification of fine spatial features including single-cell layers in the mouse brain[61]. It revealed cell zonation patterns and cellular constituents, consistent with findings observed in human post mortem brain tissue. Tissue structure dimensions did not change, indicating minimal lateral diffusion from the barcoded beads. Analysis of gene expression following traumatic brain injury revealed cell proliferation progressing to differentiation. This technique accurately reflected spatial distribution of classical neuronal and non-neuronal cell types, ~66% of the DNA-barcoding beads matched to single-cell types, while ~33% matched to two cell types. Expression values concurred with results from bulk-mRNA-seq and scRNA-seq, and average mRNA capture was consistent across all tissues[61]. Advancements have further improved spatial resolution, such as high-definition spatial transcriptomics (HDST) which employs 2 μm spatial barcoding[63] and permits histological analysis which is not possible with Slide-seq.

## Computational approaches for resolving spatial gene expression

*Integrating spatial and expression information.* It is possible to map scRNA-seq data onto *Slide-seq* data using non-negative matrix factorisation regression (NMFreg) which reconstructs *Slide-seq* expression as a combination of cell-type signatures from scRNA-seq[61]. LIGER (linked inference of genomic experimental relationships) is a computational method for spatially locating cells present in scRNA-seq data from in situ transcriptomic data, thereby increasing the resolution of the in situ data[64]. The

Harmony algorithm, which projects cells into a shared embedding in which cells group by cell type rather than dataset-specific conditions, is shown to be both efficient and accurate[65]. Another multiple datasets integration pipeline, inspired by multiple sequence alignment, utilises canonical correlation analysis (CCA) to identify anchor points across heterogeneous datasets[66]. An alternative approach is SpaOTsc, which utilises genes with spatial measurements to extrapolate the spatial properties of scRNA-seq data[67]. A different approach is found in Giotto, a user-friendly workspace that utilises cell-type-specific gene signatures to infer cell-type enrichment scores for downstream analyses with the capability for integration of spatial information[68]. Altogether, these computational approaches present timely developments to capitalise on the rapid growth of high-throughput tissue sectioning-based technologies and are summarised in Table 2. However, these methods cannot yet incorporate multiple data types, such as gene expression and intergenic methylation, in defining cell types.

*De novo spatial position prediction using only expression data.* Rather than combining two different datasets, a new class of advanced computational techniques enables the prediction of spatial information from a single gene expression dataset. *novoSpaRc* is a recent gene expression cartography technique that performs mapping based on the variation of gene expression across a tissue section, using a probabilistic optimisation technique[69]. However, the accuracy can be sub-par due to the lack of reference map, and a set of a priori marker genes with known expression patterns is desirable. In *Drosophila*, ScoMAP (Single-Cell Omics Mapping into spatial Axes using Pseudotime ordering) is another reference-free technique that spatially integrates expression data into a virtual latent space, resembling the organization of a 2D tissue[70]. At the single-cell resolution level, the CSOmap (Cellular Spatial Organization mapper) algorithm can partially reconstruct the tissue spatial organisation based on ligand-receptor interaction[71]. Altogether, these computational advances demonstrate the potential power of in silico techniques in de novo spatial mapping of expression data, but there is still room for improvement in prediction accuracy.

**High-throughput native single-cell approaches.** Recently, the emergence of high-throughput single-cell resolution 3D analysis methods has provided the spatio-temporal expression information necessary to understand embryogenesis and development. The main advantage of these techniques over the previous integrating methods, is that these do not require prior annotation, nor do they rely on tissue dissociation to obtain quantitative values.

*novoSpaRc* (Fig. 1k) is a new computational framework which allows de novo spatial reconstruction of single-cell gene expression[69]. Previous computational approaches have required a reference atlas of marker genes to assign spatial coordinates. Prior knowledge is not required for *novoSpaRc*, as this method takes single-cell transcriptome profiles, sequenced from dissociated cells as its input, then returns a virtual tissue of a chosen shape, which can be queried for the expression of all genes quantified in the data. There is a high degree of variability in how these sequenced cells may be arranged. The *novoSpaRc* hypothesis is that genes are often expressed in spatially contiguous territories, and by tracking cells exhibiting similar features, the best matching spatial arrangement of cells can be found. Spatial patterns are reconstructed with little prior knowledge, by looking for the spatial arrangement of sequenced cells in which nearby cells have transcriptional profiles that are often more similar than cells that are further apart.

**Table 2 Computational tools and associated methods for spatial transcriptomics data analysis and visualisation.**

| Tool | Underlying method | Open-source | Output | Input | Programming language | Source code |
|---|---|---|---|---|---|---|
| RNAscope[29] | Proprietary software | No | N/A | N/A | N/A | N/A |
| DistMap[50] | Distributed mapping scores | Yes | Expression patterns | Count matrix, reference in situ coordinates | R | https://github.com/rajewsky-lab/distmap |
| NMFreg[61] | Non-negative matrix factorization regression | Yes | Expression patterns | Count matrix, 2D spatial coordinates | Python | https://github.com/tudaga/NMFreg_tutorial |
| LIGER[64] | Integrative non-negative matrix factorization | Yes | Expression patterns, cell clusters | Count matrix, scMethylation, scATAC-seq | R | https://github.com/MacoskoLab/liger |
| Harmony[65] | Maximum diversity clustering, linear batch correction | Yes | Expression patterns | Count matrix, MERFISH coordinates | R | https://github.com/immunogenomics/harmony |
| SpaOTsc[67] | Structured optimal transport | Yes | Expression patterns | Count matrix, spatial coordinates, dissimilarity matrices | Python | https://github.com/zcang/SpaOTsc |
| NovoSpaRc[69] | Generalised optimal-transport | Yes | Expression patterns | Count matrix, target space image | Python | https://github.com/rajewsky-lab/novosparc |
| ScoMAP[70] | Axial information extraction via pseudotime ordering | Yes | Expression patterns | Count matrix, virtual spatial template | R | https://github.com/aertslab/ScoMAP |
| CSOmap[71] | Cellular Spatial Organization mapper | Yes | Expression patterns | Count matrix, label, ligand-receptor | MatLab | https://codeocean.com/capsule/2860903/tree/v1 |
| Giotto[68] | Binary Spatial extraction, hidden Markov random field (HMRF) model | Yes | Expression patterns | Count matrix, spatial coordinates | R, Python | https://rubd.github.io/Giotto/ |
| SPOTlight[98] | Seeded non-negative matrix factorization (NMF) regression | Yes | Expression patterns | Count matrix, spatial coordinates | R | https://github.com/MarcElosua/SPOTlight |

Analysis of newly synthesised RNA has previously been limited to the level of cell populations, as weak detection tools required a large amount of total RNA. Dynamically produced RNA is an important measure of rapid gene expression in response to stimuli, a response which is not universal across cell populations. For this reason, a single-cell resolution technique is required. New transcriptome alkylation-dependent single-cell RNA sequencing (NASC-seq)[72] is an emerging disruptive technology which monitors newly synthesised and pre-existing RNAs simultaneously (Fig. 1l), by sequencing the T–C chemical conversions in response to 4-thiouridine (4sU) labelling. This can be used to computationally separate, old and new RNA transcripts by sequence analysis, providing spatial and temporal monitoring at the single-cell level.

## Data visualisation and management

There is high demand for public repositories of spatially resolved transcriptomic data, which allow comparison and integration of multiple sources and resolutions of analysed output[73]. Progressing beyond curated databases, 3D-visualisation cell atlases modelling detailed spatial data will provide the most benefit in research and medical applications[74]. Several applications exist which allow high fidelity imaging datasets to be integrated with quantitative data for completely interactive analysis, such as Virtual Fly Brain[75], and the atlas of embryonic human hearts which combines *Spatial Transcriptomics* and scRNA-seq[60]. Results from *DistMap* were used to generate *Drosophila* Virtual Expression eXplorer (DVEX), an online, browsable database which generates virtual ISHs capable of predicting spatially resolved expression patterns in stage 6 embryos. Expression data is currently limited to binarized bins under defined thresholds, each expressing 6500–8500 genes, and cells are mapped independently with no incorporation of previously mapped cells into score refinement. Other 3D-visualisation tools such as MorphoNet[76] will assist in integrating quantitative information from different platforms or sources on the same 3D object.

## Data management standards

**Data management standards**. To promote reproducibility, standards for data storage, analysis and exchange are urgently needed, as spatial transcriptomics currently lack common guidelines for data management, compared to more mature sequencing technologies (e.g. RNA-seq, scRNA-seq)[77]. Fortunately, the majority of analysis tools in spatial transcriptomics are publicly available on GitHub (e.g. https://github.com/10XGenomics, https://github.com/SpatialTranscriptomicsResearch), paving the way for building community standards, potentially enabling integration of different spatial transcriptomics technologies to address a common biological question. Further, it has become standard practice in the research community to have preprints publicly available (e.g. BioRXiv.org, MedRXiv.org), allowing rapid dissemination of methods and results, and lessons can be learned from the systems biology standards[78,79]. This presents a great opportunity and a critical goal for the spatial transcriptomics community now that various technologies are in the developmental phase.

## Discussion and future directions

Standard techniques such as in situ hybridisation are widely used to interrogate spatial gene expression. Based on visual examination, these techniques are extremely powerful in detecting changes in gene expression patterns. While providing useful data spatially, the quantitative power of fluorescence imaging remains low[80]. Tools have continued to develop, with various improvements to the efficiency, versatility and power of these non-invasive imaging approaches[81–83]. Recent improvements in single-molecule detection rendered this technology amenable for quantitative measurements and larger throughput, forecasting future scalability to assess whole transcriptomes.

RNA sequencing and its derivative technologies have answered the need for quantitative data and are suitably high-throughput for assaying whole genomes. These methods, however, lack the acuity necessary to deliver spatial results. Combining ISH and scRNA-seq using computational integration, achieves the level of accuracy required for resolving accurate spatio-temporal gene expression pattern. Identification of spatial markers and archetypal expression patterns is the basis for these current integrative approaches. Cell types can be visualised in 3D with computational techniques, however, this is based on measurement of spatial information from 2D slides and careful sampling is required to ensure faithful recapitulation of spatial patterning. Challenges remain in scaling these methods to accommodate large tissue volumes, however, as sequencing costs drop, it is forecast that it will be feasible for these technologies to scale to whole organisms. High-resolution, imaging-based approaches require high magnification and fine sectioning necessitating long imaging times, which will protract further with scaling. Methods which employ targeted probes carry the inherent limitation of finite available fluorophores, and will be challenging to scale. Underpinning these challenges is that, as more cells and tissues are analysed with higher resolution, more data points are required, demanding higher computational power, and scalable mathematical models for future 3D visualisation and data interpretation. Spatial technologies vary in their readiness for scaled analysis, and this will prove a key determinant of the lifespan and relevancy of the technology. Further limitations remain such as the difficulties in obtaining reproducible results due to single-cell analysis variability, and the applications in heterogeneous tissues such as eye retinas and cancer tumours remain to be determined. Methods to integrate spatial measurements with scRNA-seq data are beginning to emerge[84], employing "multimodal intersection analysis" to capitalise on the strength of information about cellular identity gained from scRNA-seq and the spatial data from microarrays to highlight spatially restricted gene networks and cell enrichment. Such approaches pave the way towards establishing virtual atlases which achieve a whole-organism transcriptome with cellular resolution. Several single-cell atlasing projects are endeavouring to develop these public resources which provide a valuable source of information for many model organisms including The Human Cell Atlas[85] and single-cell zebrafish transcriptome atlas[86]. Ultimately, spatially resolved transcriptomics will be paramount in adding spatial coordinates to these single-cell atlasing projects[87].

Accessibility to spatially resolved methods will determine popularity and speed of incorporation into research. Commercialisation of novel technologies supports consistent and reliable results, and promotes competition-driven enhancements to sensitivity and efficiency. CARTANA (Sweden) offers kits and servicing for padlock-based in situ sequencing technologies, while 10X Genomics acquired Spatial Transcriptomics and their Visium Spatial Gene Expression system reduced barcode spacing, leading to an improved resolution and 55 μm microarray[88]. Ongoing modifications and novel data integration enhance utility, providing new versions of existing technologies such as Slide-seqV2 which is currently in development[89].

Several types of RNA molecules are active within the cell, engaging in diverse roles from mRNA encoding proteins, to noncoding and small RNAs regulating gene expression at the transcriptional, post-transcriptional and epigenomic levels[90]. To generate a transcriptomic map capable of capturing the complexity of gene expression, it must be possible to detect a range of RNA molecules, unrestricted by length, and have the ability to distinguish between similar isoforms. Technologies which lack sequencing capabilities, also lack the ability to discriminate

between many alternate and splice forms, and cannot faithfully predict the effect of RNA structure. Approaches like smFISH which require multi-probe binding, restrict the discovery of short sequences of RNA such as microRNA and tRNA, and limit the profiling of different isoforms. Technologies which fail to capture subtle variations in sequence length and identity, or which lack the resolution to distinguish unique isoforms of RNA, comprise the complexity of these systems, and novel insights regarding cell type and gene function may be overlooked.

Rather than supplant existing techniques, many new approaches are designed to integrate with current technologies, expanding the scope of retrieved data. Many fields of science provide valuable insights into spatial genomics, however, without a wide-ranging, collaborative effort, these assets will remain underutilised. Mass spectrometry imaging is one such avenue yet to be fully explored for its integrative potential, despite demonstrating clear benefits in applications such as proteogenomics where structural imaging is followed by RNA sequencing[91]. Computationally aligning spatially registered images with quantitative expression data will allow a unique appreciation of the complexity of tissue structure and composition. High-throughput spatial transcriptomics are becoming an essential component of both bench and bedside medicine, with an increasing demand for efficient analysis pipelines in clinical histology and pathology. The ultimate goal is to generate spatial-omics data, where a single sample may be non-destructively analysed to reveal temporal, spatial, and quantitative data simultaneously.

Integration of temporal expression datasets will enhance understanding of spatial transcriptomics. Incorrect temporal expression patterns can lead to developmental disorders or diseases like cancer. Understanding the gene expression timescale is crucial to identifying key triggers at different stages of development. For instance, single-cell studies performed at different stages of pluripotent cells differentiating into cardiomyocytes, revealed novel developmental pathways underlying lineage specification during cardiogenesis[92]. Recent reports capturing expression data in vivo by tracking transcription factor binding have illustrated the possibilities of these techniques[93], including developing predictive GRNs[94]. Integrating 3D regulatory information, such as epigenomics studies, will complete our appreciation of the GRN controlling spatio-temporal gene expression patterns. Advances in epigenomic studies to allow single-cell analysis, such single-cell ATAC-seq[95] have provided new insights into regulatory heterogeneity in complex tissues[96]. Ultimately, incorporating chromatin architecture and regulation in a 3D digital embryo will provide a comprehensive framework to uncover the GRNs executing spatio-temporal control of gene expression[97]. This promises to accurately predict the downstream effects of genetic perturbations, such as those caused by mutations resulting in developmental disorders or disease, in an immediate, automated manner.

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

## Acknowledgements

We thank the members of the Ramialison Laboratory for their support. This project is funded by the Australian Research Council Discovery Project DP190102771 to M.R. The Australian Regenerative Medicine Institute is supported by grants from the State Government of Victoria and the Australian Government.

## Author contributions

L.N.W., L.G.M. and M.R. designed the study. L.N.W. wrote the manuscript with contributions from M.R., H.T.N. and L.G.M.

## Competing interests

The authors declare no competing interests.
