## [Peer Review File · Communications Biology]

Reviewers' comments:

Reviewer #1 (Remarks to the Author):

I'm happy to see this review of methods allowing spatial mapping of gene expression patterns and how can be linked to bulk measurements. I'm glad to see the authors undertook this task given the many techniques and methods needed a review.

1. In the introduction, it will be good to link this review to previous efforts to create 3D atlases of gene expression using bulk methods and the associated computational approaches and applications thereof. For example, the Allen Institute for Brain Sciences have created a comprehensive in situ hybridization-based atlas of the mouse brain and followed up with human and macaque atlases. These have led to significant advancement of our understanding of brain function despite the lack of quantitative data or the low resolution.
2. The paper focuses mainly on mRNA and other species of RNA molecules are rarely mentioned. A discussion of these methods' ability to profile different isoforms will be very valuable for the reader. For instance, it is mentioned that smFISH requires using multiple probes which limits its application to short mRNAs. The same reason also limits its ability to profile different isoforms.
3. Table 1 can be enhanced by adding columns to compare the methods in terms of sensitivity, ease of tissue preparation, ease of setup, commercial availability, etc. I find the "Throughput" column a bit confusing because the authors use a mix of categorical values and numerical values. I suggest unifying this for better comparisons. And I urge the authors to report range of numbers per method.
4. Line 224: I don't understand how this FACS example fits with the rest of the paragraph where the combination of spatial information from in situ studies, with the quantitative data obtained from RNA-seq is discussed. Please clarify.
5. While there are several methods available to integrate scRNA-seq and spatial data, the authors only mentioned LIGER in association with SlideSeq (line 375). I suggest discussing these methods briefly in a separate paragraph to pinpoint the added value of the integration and the current limitation/challenges.
6. The authors pinpointed challenges associated with visualization aspects of this data. While important, I believe a bigger challenge lies in the lack of standards in storing and sharing spatial transcriptomic data, compared to RNA-seq data. This is perhaps something the community can address now while these methods are still in the development phase.
7. In the outlook section, perhaps it is good to pinpoint that we should learn from other domains. For instance, imaging mass spectroscopy. They have been ahead of RNA techniques. One thing we foresee is that we will need to integrate these spatial transcriptomic data with microscopy data (using image registration/alignment) to bring these technologies to clinical practice where thousands of pathology slides are scanned routinely.

Reviewer #2 (Remarks to the Author):

The author gave a comprehensive review of the single-cell spatial genomics technology.

Commons:

1. The author mentioned "quantitative accuracy" multiple times (e.g., line 20) and treated it as one of the major drawbacks of FISH tech compared to scRNA-seq. However, the definition of "quantitative accuracy" is not claimed in the article. The authors should give the definition/background first since scRNA-seq is known to be highly noisy so the readers might feel confused that why FISH tech is even worse than scRNA-seq in the aspect of "accuracy".

2. The author introduced Seurat (line 231) and other methods (Distmap, etc.) for integrating scRNA-seq data with in situ. However, Seurat keeps actively updating since first published in 2015. For instance, in the paper "Comprehensive integration of single-cell data" (Cell, 2019), the authors of Seurat proposed a new single cell data integration algorithm, which showed nice results in integrating scRNA-seq data with FISH data. Another algorithm, Harmony (Fast, sensitive and accurate integration of single-cell data with Harmony, Nature Methods, 2019), also achieved good integration of scRNA-seq data and FISH data. Therefore, this article should try to include more state-of-art methods in scRNA-seq / FISH data integration.

3. Table 1 gives a summary of all methods in this article. It would be beneficial if this table can include some pros/cons or a very brief description of the strength of each technique. With such information, this summary table would be a better guide for method selection in practice. Also, why does only STARMap show the throughput as "~1000" while all the others show "low/med/high"? It would be better to keep it consistent, and the author may consider all use numbers instead of just "low/med/high" category.

4. Figure 1 gives a summary of the primary mechanism of each technology. This figure needs more annotation since the current version is not intuitive, with lots of different symbols that lack annotation.

Minor issues:

1. several "in situ" are not italic.

2. Figure 1 uses "bRNA-seq" as the abbreviation of bulk RNA-seq. This abbreviation is not commonly used.

Reviewer #3 (Remarks to the Author):

In this review, Waylen and colleagues comprehensively review emerging high-throughput technologies for spatially-resolved gene expression measurements. The review is overall very well organized, and highlights both imaging based (such as MERFISH, seqFISH) and next-generation sequencing based spatial transcriptomics technologies (Slide-seq, Spatial Transcriptomics (ST), etc). For each technology discussed the authors concisely describe the fundamentals of how the method works, its applications, and also its limitations. We believe that this is an excellently written review that concisely documents the development of tools driving the burgeoning field of spatial transcriptomics.

Although the authors do cite studies demonstrating the integration of scRNA-seq with spatial transcriptomic data modalities, the authors may wish to make this more of a focal point in the 'Future Directions' section of the review. While this is certainly cited and discussed (Rodrigues et al 2019 for Slide-seq, Asp et al 2019 for ST), the review would benefit from a few sentences discussing how these spatial gene expression studies fit into the growing compendium of scRNA-seq 'atlasing' projects.

While the challenge of integrating spatial measurements and (sc)RNA-seq data is highlighted (lines 447) particularly for heterogeneous tissue types such as the eye retina and tumors (lines 452), the authors may wish to consider citing other published methods that have been published for integrating spatial and RNA-seq data modalities (such as Moncada et al Nature Biotechnology, 2020).

In addition to these points, we have the following comments:

- Lines 153-155: while the numbers mentioned are relevant to the original MERFISH technology,

MERFISH+ (Xia et al PNAS 2019, Pubmed ID 31501331) detects transcripts from ~10,000 genes at cellular resolution.

- Where the 'Spatial Transcriptomics' technology is discussed (starting on line 343), it should be highlighted that this particular microarray-based methodology provides histological information from the same tissue section used for the Spatial Transcriptomics assay. This is not the case for any of the other NGS based methods discussed.
- The information regarding the Spatial Transcriptomics technology is outdated - since the acquisition of the technology by 10X Genomics, the assay boasts increased spatial resolution: from 100 μm to 55 μm features, in addition to microarray spots being printed closer together. The authors may wish to consider mentioning this in the text and/or in Table 1.
- The authors do not discuss the size of the tissue appropriate for the microarray based Spatial Transcriptomics nor the Slide-seq assays. For Spatial Transcriptomics, the barcoded array is about ~ 6mm x 6mm, while Slide-seq uses a ~5 mm (diameter) circular bead puck. These details should be noted in the text.
- The authors do not discuss the High Definition Spatial Transcriptomics method (Vickovic et al Nature Methods 2019, Pubmed ID 31501547) which uses 2 μm features.
- On line 377, the authors write "Cell types can be visualised in 3D with computational programming however this is based on measurement of spatial information from 2D slides and careful sampling is required to ensure faithful recapitulation of spatial patterning." This is written in the 'Slide-seq' section. Why? In the original Slide-seq manuscript the authors do not describe 3D tissue reconstruction from the data. This statement seems generalizable to any of the technologies described in this review and should be moved to the 'Future Directions' section.
- Furthermore, on line 380, the authors go on to write "Currently Slide-seq is capable of scaling to sequence large tissue volumes, and as sequencing costs drop, is forecast to be able to scale whole organisms." This statement isn't very well supported, given that the authors make no mention of the tissue size restrictions of the Slide-seq assay. This statement is also generalizable to all technologies. These comments may be better fit in the 'Future Directions' section of the review.
- On page 9 line 270, GRN is not defined (gene regulatory network?)

Response to Reviewers:

We thank the reviewers for the positive and constructive feedback and for their support for the need for this review in the field of spatially resolved transcriptomics. We have addressed all their suggestions by substantially revising the manuscript and adding novel sections. Changes to the original manuscript have been highlighted in blue.

Reviewer #1

single-cell genomics/transcriptomics, computational methods

I'm happy to see this review of methods allowing spatial mapping of gene expression patterns and how can be linked to bulk measurements. I'm glad to see the authors undertook this task given the many techniques and methods needed a review.

Question #1:

In the introduction, it will be good to link this review to previous efforts to create 3D atlases of gene expression using bulk methods and the associated computational approaches and applications thereof. For example, the Allen Institute for Brain Sciences have created a comprehensive in situ hybridization-based atlas of the mouse brain and followed up with human and macaque atlases. These have led to significant advancement of our understanding of brain function despite the lack of quantitative data or the low resolution.

Response:

We thank the Reviewer for this suggestion and have included a new paragraph discussing the previous efforts to generate 3D gene expression atlases and their subsequent updates. **Page 4, lines 74-83.** We very much welcomed the suggestion to explore the computational approaches and therefore have included a new section ‘*Computational approaches for resolving spatial gene expression*’ which discusses state-of-the-art computational approaches and their applications, in spatial gene expression technologies. **Pages 13-14, lines 398-432.**

Page 4, lines 74-83:

Previous applications that have integrated bulk and computational methods with *in situ* hybridisation gene expression data include the mouse brain atlas constructed by the Allen Institute for Brain Sciences (14), later accompanied by human and macaque atlases (15, 16). Despite challenging low resolution and lack of quantitative data, these atlases significantly advanced the current understanding of brain structure and function. More recently, the Allen Mouse Brain Common Coordinate Framework (CCFv3) (17) sampled 1,675 mouse brains at 10 μm voxel resolution, integrating multiple datasets in 3D. Virtual atlases are a valuable source of information for many model organisms including *C. elegans* (Wormbase) (18), frogs (Xenbase) (19), *Drosophila* (20), and zebrafish (21), and publicly available, browsable data supports myriad fields of research.

Pages 13-14, lines 398-432:

Computational approaches for resolving spatial gene expression

Integrating spatial and expression information. It is possible to map scRNA-seq data onto Slide-seq data using non-negative matrix factorisation regression (NMFreg) which reconstructs Slide-seq expression as a combination of cell type signatures from scRNA-seq (62). LIGER (linked inference of genomic experimental relationships) is a computational method for spatially locating cells present in scRNA-seq data from *in situ* transcriptomic data, thereby increasing the resolution of the *in situ* data (65). The Harmony algorithm, which projects cells into a shared embedding in which cells group by cell type rather than dataset-specific conditions, is shown to be both efficient and accurate (66). Another multiple datasets integration pipeline, inspired by multiple sequence alignment, utilises canonical correlation analysis (CCA) to identify anchor points across heterogeneous datasets (67). An alternative approach is SpaOTsc, which utilises genes with spatial measurements to extrapolate the spatial properties of scRNA-seq data (68). A different approach is found in Giotto, a user-friendly workspace that integrates spatial expression data and cell type specific gene signatures to infer cell type enrichment scores for downstream analyses (69). Altogether, these computational approaches present timely developments to capitalise on the rapid growth of high-throughput tissue sectioning-based technologies and are summarised in Table 2. However, these methods are not yet capable of incorporating multiple data types, such as gene expression and intergenic methylation, in the definition of cell types.

de novo spatial position prediction using only expression data. Rather than combining two different datasets, a new class of advanced computational techniques enables the prediction of spatial information from a single gene expression dataset. NovoSpaRc is a recent gene expression cartography technique that performs mapping based on the variation of gene expression across a tissue section, using a probabilistic optimisation technique (70). However, the accuracy can be sub-par due to the lack of reference map, and a set of *a priori* marker genes with known expression patterns is desirable. In *Drosophila*, ScoMAP (Single-Cell Omics Mapping into spatial Axes using Pseudotime ordering) is another reference-free technique that spatially integrates expression data into a virtual latent space, resembling the organization of a 2D tissue (71). At the single-cell resolution level, the CSOmap (Cellular Spatial Organization mapper) algorithm can partially reconstruct the tissue spatial organisation based on ligand-receptor interaction (72). Altogether, these computational advances demonstrate the potential power of *in silico* techniques in *de novo* spatial mapping of expression data, but there is still room for improvement in prediction accuracy.

Question #2

The paper focuses mainly on mRNA and other species of RNA molecules are rarely mentioned. A discussion of these method's ability to profile different isoforms will be very valuable for the reader. For instance, it is mentioned that smFISH requires using multiple probes which limits its application to short mRNAs. The same reason also limits its ability to profile different isoforms.

Response:

We agree with the Reviewer's suggestions and have included a novel discussion on the capability of these methods to investigate the length and identity of RNA molecules.

Page 18, lines 542-554:

Several types of RNA molecules are active within the cell, and they engage in diverse roles from mRNA encoding proteins to non-coding and small RNA regulating gene expression at the transcriptional, post-transcriptional and epigenomic levels (89). To generate a transcriptomic map able to capture the true complexity of gene expression, it must be possible to detect a range of RNA molecules, unrestricted by length, and have the ability to distinguish between similar isoforms. Technologies which lack sequencing capabilities, also lack the

ability to discriminate between many alternate and splice forms, and cannot faithfully predict the effect of RNA structure. Approaches which require multiple probes binding such as smFISH, restrict the discovery of short sequences of RNA such as microRNA and tRNA, and limit its ability to profile different isoforms. Technologies which fail to capture subtle variations in sequence length and identity, or which lack the resolution to distinguish unique isoforms of RNA, comprise the complexity of these systems, and novel insights and information regarding cell type and gene function may be overlooked.

Question #3:

Table 1 can be enhanced by adding columns to compare the methods in terms of sensitivity, ease of tissue preparation, ease of setup, commercial availability, etc. I find the "Throughput" column a bit confusing because the authors use a mix of categorical values and numerical values. I suggest unifying this for better comparisons. And I urge the authors to report range of numbers per method.

Response:

We have incorporated the Reviewer's suggestions into Table 1. The Reviewer is correct that previously "Throughput" column referred to a combination of number of genes and cells. This has now been clarified by separating the number of cells (referring to a bracket of number of cells assayed at the same time) and by the addition of number of genes column (which are now numerical values). In addition, a new "Coverage" column was added and "Spatial Resolution" has been revised to include more accuracy. A new row was added to cover a new technology HDST. We found that "ease of tissue preparation, ease of setup" is challenging to discuss as these parameters are fluid and context-dependant. However, commercialisation is an avenue to standardise the use of the technologies, therefore we included a paragraph on commercial availability in the revised Discussion section. **Pages 17-18. Lines 532-540.**

Table 1. Features of current methods for capturing spatial gene expression.

Methodology	Coverage	Number of Genes	No. of cells ¹	Spatial Resolution
in situ hybridisation (20)	Targeted	3	Low	Tissue
RNA-scope (27)	Targeted	12	Low	Cellular
ClampFish (28)	Targeted	3	Low	Subcellular
smFISH (31)	Targeted	3	Low	Subcellular
osmFISH (30)	Targeted	1-33+	Low	Subcellular
MERFISH (33)	Targeted	10,000	Medium	Subcellular
DNA Microscopy (38)	Targeted	24	Low	Cellular
seqFISH+ (39)	Targeted	10,000	High	Subcellular
DistMap (49)	Targeted	8000+	High	Cellular
STARMap (50)	Targeted	1020+	High	Cellular
Tomo-seq (51)	Transcriptome-wide	Whole transcriptome	High	Cellular
Geo-seq (55)	Transcriptome-wide	Whole transcriptome	High	Cellular
Spatial Transcriptomics /10X Visium (58)	Transcriptome-wide	Whole transcriptome	Medium	100µm/55µm

¹ Number of Cells; Low 0-100, Medium 100-1000, High 1000-10,000+

Slide-seq (60)	Transcriptome-wide	Whole transcriptome	Medium	10µm
HDST	Transcriptome-wide	Whole transcriptome	High	2µm
novoSpaRc (68)	Transcriptome-wide	Whole transcriptome	High	Cellular
NASC-seq (71)	Transcriptome-wide	Whole transcriptome	High	Cellular

Pages 17-18. Lines 532-540:

Accessibility to these spatially resolved methods will determine popularity and speed of incorporation into new research. Commercialisation of novel technologies not only supports consistency and reliability of results, but promotes competition-driven enhancements to sensitivity and efficiency. CARTANA (Sweden) offers kits and servicing for padlock-based *in situ* sequencing technologies, while 10X Genomics acquired Spatial Transcriptomics and their Visium Spatial Gene Expression system reduced barcode spacing leading to an improved resolution and 55µm microarray (87). Ongoing modifications and novel integration of data will enhance utility, providing new versions of existing technologies such as Slide-seqV2 which is currently in development (88).

Question #4:

Line 224: I don't understand how this FACS example fits with the rest of the paragraph where the combination of spatial information from in situ studies, with the quantitative data obtained from RNA-seq is discussed. Please clarify.

Response:

We apologise for the confusion; indeed the Reviewer is right, this study did not utilise ISH, but investigated different spatial zones of the infarcted heart, we have clarified this in the manuscript as follows.

Page 9, Lines 240-243:

Recent experiments to combine spatial information, with the quantitative data obtained from RNA-seq have been used to query the transcriptional changes in different regions of the cardiac infarct zone following heart injury in adults and neonates, providing type specific quantitative expression data (45).

Question #5:

While there are several methods available to integrate scRNA-seq and spatial data, the authors only mentioned LIGER in association with SlideSeq (line 375). I suggest discussing these methods briefly in a separate paragraph to pinpoint the added value of the integration and the current limitation/challenges.

Response:

We have followed the Reviewer’s suggestion and introduced a novel section dedicated to discussing the advantages and limitations of a range of current data integration methods which include (in addition to LIGER), SpaOTsc, Giotto, ScoMAP, CSOmap and the Harmony algorithm. **Pages 13-14. Lines 398-432, see also Response to Question #1.** In this section entitled “*Computational approaches for resolving spatial gene expression*” we have articulated this discussion by dividing the computational tools into two classes: 1) Integrating spatial and expression information and 2) *de novo* spatial position prediction using only expression data. In addition, we have summarised the computational tools and their associated methods in a new Table 2.

Table 2. Computational tools and associated methods for spatial transcriptomics data analysis and visualisation.

Tools	Underlying Methods	Open-source	Programming languages	Source code
RNAScope (29)	proprietary software	No	N/A	N/A
DistMap (51)	distributed mapping scores	Yes	R	https://github.com/rajewsky-lab/distmap
NMFreg (62)	non-negative matrix	Yes	Python	https://github.com/tudag

	factorization regression				a/NMFreg_tutorial
LIGER (65)	integrative non-negative matrix factorization	Yes	R		https://github.com/MacoskoLab/liger
Harmony (66)	maximum diversity clustering, linear batch correction	Yes	R		https://github.com/immunogenomics/harmony
SpaOTsc (68)	structured optimal transport	Yes	Python		https://github.com/zcang/SpaOTsc
NovoSpaRc (70)	generalised optimal-transport	Yes	Python		https://github.com/rajewsky-lab/novosparc
ScoMAP (71)	axial information extraction via pseudotime ordering	Yes	R		https://github.com/aertslab/ScoMAP
CSOmap (72)	Cellular Spatial Organization mapper	Yes	MatLab		https://codeocean.com/capsule/2860903/tree/v1
Giotto (69)	binary spatial extraction, hidden Markov random field (HMRF) model	Yes	R, Python		https://rubd.github.io/Giotto/
SPOTlight (97)	seeded non-negative matrix factorization (NMF) regression	Yes	R		https://github.com/MarcElosua/SPOTlight

Question #6:

The authors pinpointed challenges associated with visualization aspects of this data. While important, I believe a bigger challenge lies in the lack of standards in storing and sharing spatial transcriptomic data, compared to RNA-seq data. This is perhaps something the community can address now while these methods are still in the development phase.

Response:

This is indeed an essential discussion therefore we have included a novel section on Data Visualisation and Management.

Page 16. Lines 485-498.

Data Visualisation and Management

[...]

Data management standards. To promote reproducibility, spatial transcriptomics urgently needs standards for data storage, analyses, and exchange. Spatial transcriptomics currently lacks common standards and guidelines for data management, as compared to the more mature sequencing technologies (e.g. RNA-seq, scRNA-seq) (76). Fortunately, the majority of analysis tools in spatial transcriptomics are publicly available on GitHub (e.g. <https://github.com/10XGenomics>, <https://github.com/SpatialTranscriptomicsResearch>), which helps pave the way for building community standards, potentially enabling integration of different spatial transcriptomics technologies to address a common biological question. Lessons can be learned from the systems biology standards (77, 78), which can help accelerate the development of standards in spatial transcriptomics. Altogether, this presents a great opportunity and a critical goal for the spatial transcriptomics community now that various technologies are in the development phase.

Question #7:

In the outlook section, perhaps it is good to pinpoint that we should learn from other domains. For instance, imaging mass spectroscopy. They have been ahead of RNA techniques. One thing we foresee is that we will need to integrate these spatial transcriptomic data with microscopy data (using image registration/alignment) to bring these technologies to clinical practice where thousands of pathology slides are scanned routinely.

Response:

We thank the Reviewer for this great suggestion, and have added the following paragraph accordingly.

Pages 18. Lines 555-566.

Many fields of science provide valuable insights into spatial genomics, however without a wide-ranging, collaborative effort, these assets will remain underutilised. Mass spectrometry imaging is one such avenue yet to be fully explored for its integrative potential, despite demonstrating clear benefits in applications such as proteogenomics where structural imaging is followed by RNA-sequencing (80). Computationally aligning spatially registered images with quantitative data will allow a unique appreciation of the complexity of tissue structure and composition. Beyond the research scope, there is an increasing demand for efficient analysis pipelines for clinical requirements in histology and pathology. High-throughput spatial transcriptomics are becoming an essential component of both bench and bedside medicine. The ultimate goal in this field of discovery is to be able to generate spatial-omics data, where a single sample may be non-destructively analysed to reveal temporal, spatial, and quantitative data simultaneously.

Reviewer #2

single-cell RNA-seq, bioinformatics, statistics

The author gave a comprehensive review of the single-cell spatial genomics technology.

Question #8:

The author mentioned "quantitative accuracy" multiple times (e.g., line 20) and treated it as one of the major drawbacks of FISH tech compared to scRNA-seq. However, the definition of "quantitative accuracy" is not claimed in the article. The authors should give the definition/background first since scRNA-seq is known to be highly noisy so the readers might feel confused that why FISH tech is even worse than scRNA-seq in the aspect of "accuracy".

Response:

We agree with the Reviewer that a meaningfully quantitative method requires sufficiently high signal-to-noise ratio for comparison across different experimental conditions in a

reproducible manner. In a FISH experiment context, the expression levels are subject to imaging parameters and therefore not easily or objectively measured. Hence, we have described the lack of “quantitative accuracy” as a major drawback of FISH. We have clarified the intended meaning of “quantitative accuracy” by replacing this term where appropriate with more transparent language, and have added the following paragraph in the manuscript in **Page 6. Lines 155-158.**

Replaced **Page 1. Line 23:** “quantitative accuracy” to “quantification”

Replaced **Page 17. Line 510:** “quantitative accuracy” to “quantitative data”

Page 6. Lines 155-158:

Where sc-RNA-seq is subject to technical and biological variations, FISH approaches provide a clear visualisation of gene expression, but lack the numerical evaluation of expression provided by sequencing and rely on making comparative evaluations for quantification.

Question #9:

The author introduced Seurat (line 231) and other methods (Distmap, etc.) for integrating scRNA-seq data with in situ. However, Seurat keeps actively updating since first published in 2015. For instance, in the paper “Comprehensive integration of single-cell data” (Cell, 2019), the authors of Seurat proposed a new single cell data integration algorithm, which showed nice results in integrating scRNA-seq data with FISH data. Another algorithm, Harmony (Fast, sensitive and accurate integration of single-cell data with Harmony, Nature Methods, 2019), also achieved good integration of scRNA-seq data and FISH data. Therefore, this article should try to include more state-of-art methods in scRNA-seq / FISH data integration.

Response:

We agree with the reviewer, given that this is a rapidly evolving field, we have endeavoured to capture the earliest and latest instances. We have now incorporated some recent algorithms, in addition to Seurat and Distmap, for integrating scRNA-seq and FISH data (SpaOTsc, CCA, Giotto, LIGER). CCA (used in Cell, 2019) and Harmony (Nature Methods, 2019) have now been extensively reviewed and discussed. **Table 2 & Pages 13-14. Lines 398-432.** Additional

computational tools have also been summarised in a new **Table 2**. For further details please also see the response to **Questions 1 and 5**.

Question #10:

Table 1 gives a summary of all methods in this article. It would be beneficial if this table can include some pros/cons or a very brief description of the strength of each technique. With such information, this summary table would be a better guide for method selection in practice. Also, why does only STARMap show the throughput as "~1000" while all the others show "low/med/high"? It would be better to keep it consistent, and the author may consider all use numbers instead of just "low/med/high" category.

Response:

The pros and cons have been discussed in the text in detail, as the technologies are constantly developing and also the pros and cons of the use of the technologies might depend on the laboratory set up and resources. We added new columns in **Table 1** to improve the accuracy of the description of the methods, and this will assist users in making better informed decisions about the technology best fit for purpose. The “*Number of cells*” column now includes numerical ranges. See also response to **Question 3**.

Question #11:

Figure 1 gives a summary of the primary mechanism of each technology. This figure needs more annotation since the current version is not intuitive, with lots of different symbols that lack annotation.

Response:

We have expanded the legend of Figure 1 accordingly to describe in further details the mechanism of each technology, including a description of the symbols used. In addition, we have edited the figure to reflect the latest available version of the Spatial Transcriptomics technology which is now High Definition Spatial Transcriptomics.

Figure 1. Principles of current methods for capturing spatial gene expression.

Schematic overview of methods based on imaging profiling of the entire specimen **(a)** *in situ* hybridisation/fluorescence staining where bound probes reveal spatial expression via fluorescent dyes **(b)** Digital Spatial Profiling where digital barcodes allow multiplexed spatial profiling **(c)** DNA Microscopy where chemical reactions permit spatial DNA imaging **(d)** seqFISH+ where accurate fluorescent imaging is performed sequentially to improve throughput and generate spatial atlases *in situ* **(e)** DistMap where Drop-seq technology integrates scRNA-seq data and ISH imaging to reveal spatial gene expression **(f)** STARmap where genes are sequenced *in situ* using padlock amplification **(g)** Tomo-seq where cryogenic tissue sections are individually analysed by bulk RNA-seq and spatial data triangulated in three axes **(h)** Geo-seq where cryogenic tissue samples are obtained through laser capture microdissection and analysed through bulk RNA-seq with results spatially mapped **(i)** High Definition Spatial Transcriptomics where cDNA synthesis is performed *in situ* and spatially barcoded prior to RNA-seq **(j)** Slide-seq where mRNA is barcoded *in situ* and spatially indexed by SOLiD **(k)** novoSpaRc where scRNA-seq is digitally profiled to virtually reconstruct the tissue **(l)** NASC-seq where 4sU labelling identifies temporal and spatial features of single cell data. Abbreviations: single-cell RNA-seq (scRNA-seq), *in situ* hybridisation (ISH), sequencing by oligonucleotide ligation and detection (SOLiD).

Question #12:

Several “in situ” are not italic.

Figure 1 uses “bRNA-seq” as the abbreviation of bulk RNA-seq. This abbreviation is not commonly used.

Response:

We thank the Reviewer for highlighting this, we have corrected these instances accordingly throughout the manuscript. **Page 6-7. Lines 165 and 170.** The abbreviation bRNA-seq is no longer used in **Figure 1**, and the phrase ‘bulk RNA-seq’ is used.

Reviewer #3

computational biology, single-cell transcriptomics

In this review, Waylen and colleagues comprehensively review emerging high-throughput technologies for spatially-resolved gene expression measurements. The review is overall very well organized, and highlights both imaging based (such as MERFISH, seqFISH) and next-generation sequencing based spatial transcriptomics technologies (Slide-seq, Spatial Transcriptomics (ST), etc). For each technology discussed the authors concisely describe the fundamentals of how the method works, its applications, and also its limitations. We believe that this is an excellently written review that concisely documents the development of tools driving the burgeoning field of spatial transcriptomics.

Question #13:

Although the authors do cite studies demonstrating the integration of scRNA-seq with spatial transcriptomic data modalities, the authors may wish to make this more of a focal point in the 'Future Directions' section of the review. While this is certainly cited and discussed (Rodrigues et al 2019 for Slide-seq, Asp et al 2019 for ST), the review would benefit from a few sentences discussing how these spatial gene expression studies fit into the growing compendium of scRNA-seq 'atlasing' projects.

Response:

We thank the Reviewer for this suggestion, and have expanded the corresponding discussion.

Page 17. Lines 527 and 532:

Such approaches are paving the way towards the establishment of virtual atlases in order to achieve a whole-organism transcriptome at a cellular resolution. Several single cell atlas projects are currently endeavouring to develop these public resources. For many model organisms, these virtual atlases are a valuable source of information including The Human Cell Atlas (86) and single-cell zebrafish transcriptome atlas (87). Ultimately, spatially resolved transcriptomics will be paramount in adding spatial coordinates to these single cell atlas projects (88).

Question #14:

While the challenge of integrating spatial measurements and (sc)RNA-seq data is highlighted (lines 447) particularly for heterogeneous tissue types such as the eye retina and tumors (lines 452), the authors may wish to consider citing other published methods that have been published for integrating spatial and RNA-seq data modalities (such as Moncada et al Nature Biotechnology, 2020).

Response:

We thank the reviewer for identifying this recent paper, and note that this area of data integration is rapidly developing, therefore we have also included a novel section on the development of novel computational methods to integrate spatial and RNAseq datasets (including single-cell). See response to Reviewer #1, Questions 1 and 5 for the new Section.

Pages 13-14. Lines 398-432.

Additionally, we have also referred to these new technologies in the revised Discussion section.

Page 17. Lines 523 and 526:

Methods to integrate spatial measurements with scRNA-seq data are beginning to emerge (79), employing “multimodal intersection analysis” to capitalise on the strength of information about cellular identity gained from scRNA-seq and the spatial data from microarrays to highlight spatially restricted gene networks and cell enrichment.

Question #15:

Lines 153-155: while the numbers mentioned are relevant to the original MERFISH technology, MERFISH+ (Xia et al PNAS 2019, Pubmed ID 31501331) detects transcripts from ~10,000 genes at cellular resolution.

Response:

We thank the reviewer for highlighting this, we have amended it accordingly. Table 1 has been modified accordingly where “*Throughout*” column has been replaced by “*Number of genes*” and “*Number of cells*” and this section now reads as:

Page 7. Lines 171-173:

This approach generates *in situ* transcriptomic analyses and detects >1000 RNA species, and provides error correction for >100. Further improvements to the MERFISH gene throughput allow simultaneous imaging of ~10,000 genes at a detection efficiency of ~80% (34). While providing a high standard of sub-cellular resolution, the necessity for cell-dissociation challenges the spatial power of this approach.

Question #16:

Where the ‘Spatial Transcriptomics’ technology is discussed (starting on line 343), it should be highlighted that this particular microarray-based methodology provides histological information from the same tissue section used for the Spatial Transcriptomics assay. This is not the case for any of the other NGS based methods discussed.

Response:

We have further emphasised this advantage of Spatial transcriptomics, and updated the text accordingly.

Page 12. Lines 366-368:

The ability to generate histological data which precisely complements the tissue section involved in quantitative analysis is unique to *Spatial transcriptomics* among other NGS methods.

Question #17:

The information regarding the Spatial Transcriptomics technology is outdated - since the acquisition of the technology by 10X Genomics, the assay boasts increased spatial resolution: from 100 μm to 55 μm features, in addition to microarray spots being printed closer together. The authors may wish to consider mentioning this in the text and/or in Table 1.

Response:

We thank the reviewer for highlighting these recent developments and have updated the text and Table 1 accordingly.

Page 18. Lines 534-542.

Accessibility to these spatially resolved methods will determine popularity and speed of incorporation into new research. Commercialisation of novel technologies not only supports consistency and reliability of results, but promotes competition-driven enhancements to sensitivity and efficiency. CARTANA offers kits and servicing for padlock-based *in situ* sequencing technologies, while 10X Genomics acquired Spatial transcriptomics and their 10X Visium system reduced barcode spacing leading to an improved resolution and 55µm microarray (80). Ongoing modifications and novel integration of data will enhance utility, providing new versions of existing technologies such as Slide-seqV2 which is currently in development (81).

Question #18:

The authors do not discuss the size of the tissue appropriate for the microarray based Spatial Transcriptomics nor the Slide-seq assays. For Spatial Transcriptomics, the barcoded array is about ~ 6mm x 6mm, while Slide-seq uses a ~5 mm (diameter) circular bead puck. These details should be noted in the text.

Response:

We thank the reviewer for this suggestion, and have included these pertinent details in the text. From the Supplemental Methods from Rodriques *et al* Science 2019 (62), we obtained the Slide-seq diameter as 3mm: “Each 3mm puck presented in this manuscript consists of roughly 70,000 beads”.

Page 12. Lines 360-363:

Spatial transcriptomics (Fig. 1i) is the least biased method to capture spatial gene expression from sections. cDNA is directly synthesised from fixed tissue sections on 6.2mm x 6.6mm pucks and is concomitantly labelled with molecular barcodes which record the spatial location of the transcript within the section (60).

Page 13. Lines 382-385:

Slide-seq (Fig. 1j) alleviates this limited resolution, and uses DNA-barcoding beads bound to 3mm slides and exposed to fresh tissue sections releasing mRNA from which barcode

sequences can be determined with sequencing by oligonucleotide ligation and detection (SOLiD) (62).

Question #19:

The authors do not discuss the High Definition Spatial Transcriptomics method (Vickovic et al Nature Methods 2019, Pubmed ID 31501547) which uses 2 µm features.

Response:

We thank the Reviewer for highlighting this, this technology is now included.

Page 13. Lines 394-396:

Advancements have further improved spatial resolution, such as High-Definition Spatial Transcriptomics (HDST) which employs 2 µm spatial barcoding (62) and permits histological analysis which is not possible with Slide-seq.

Question #20:

On line 377, the authors write “Cell types can be visualised in 3D with computational programming however this is based on measurement of spatial information from 2D slides and careful sampling is required to ensure faithful recapitulation of spatial patterning.” This is written in the ‘Slide-seq’ section. Why? In the original Slide-seq manuscript the authors do not describe 3D tissue reconstruction from the data. This statement seems generalizable to any of the technologies described in this review and should be moved to the ‘Future Directions’ section.

Response:

We agree with the Reviewer and we have moved this accordingly.

Page 17. Lines 515-518:

Identification of spatial markers and archetypal expression patterns is the basis for these current integrative approaches. Cell types can be visualised in 3D with computational

programming however this is based on measurement of spatial information from 2D slides and careful sampling is required to ensure faithful recapitulation of spatial patterning.

Question #21:

Furthermore, on line 380, the authors go on to write “Currently Slide-seq is capable of scaling to sequence large tissue volumes, and as sequencing costs drop, is forecast to be able to scale whole organisms.” This statement isn’t very well supported, given that the authors make no mention of the tissue size restrictions of the Slide-seq assay. This statement is also generalizable to all technologies. These comments may be better fit in the ‘Future Directions’ section of the review.

Response:

We agree with the Reviewer and have amended the future directions section which now includes the following.

Page 17. Lines 518-520:

While challenges remain in scaling these methods to accommodate large tissue volumes, as sequencing costs drop, it is forecast that these technologies will be feasible to scale to whole organisms.

Question #22:

On page 9 line 270, GRN is not defined (gene regulatory network?)

Response:

The introduction now includes the following definition.

Page 3. Line 44:

“Our body plan relies on spatial expression, achieved by correct deployment of a developmental gene regulatory network (GRN) where the location, timing, and level of developmental gene expression are crucial.”

REVIEWERS' COMMENTS:

Reviewer #1 (Remarks to the Author):

I would like to thank the authors for sufficiently addressing all the comments. I have no further remarks.

Reviewer #2 (Remarks to the Author):

The authors did a thorough revision based on the reviewers' comments. I only have the following additional comments.

The term "virtual atlases" was not defined.

In the description of Giotto, it is unclear how spatial information is used "to infer cell type enrichment scores for downstream analyses"

For "Computational approaches for resolving spatial gene expression," it would be useful to add a table that summarizes the input and output of each method.

I suggest English editing, as there are many places where a singular countable noun was used without an article. Also, several sentences are long and difficult to read. Other examples include "development phase" should be "developmental phase"

This sentence is still not clear: "Where sc-RNA-seq is subject to technical and biological variations, FISH approaches provide a clear visualisation of gene expression, but lack the numerical evaluation of expression provided by sequencing and rely on making comparative evaluations for quantification." I don't think it is fair to say that FISH lacks numerical evaluation because numerical values can be extracted from images. Maybe the authors can change it to "FISH approaches rely on downstream image analysis to extract numerical measurements of gene expression." Meanwhile, scRNA-seq also requires reads filtering, mapping, and quality control to be converted into numerical measurements. Therefore, I do not think the authors should simply say that FISH is less quantitative than scRNA-seq.

The response to Question 16 is unsatisfactory. The reviewer meant to emphasize that microarray-based spatial transcriptomics assay is different from NGS. The response said that this assay is a type of NGS, contradicting to the reviewer's comment.

The response to Question 21 is unsatisfactory. The reviewer meant that the generalization would not be trivial. However, the rewritten sentence still reads that the generalization is easy after the sequencing cost drops.

Reviewer #3 (Remarks to the Author):

The authors have addressed all of my concerns and I am happy to support publication at this time.

Author Response

Reviewer #2:

Comment #1:

The term "virtual atlases" was not defined.

Response:

Where the term "virtual atlases" is first used, the description has been elaborated.

Page 3. Lines 65-69:

"Virtual atlases are publicly available, browsable cell reference maps where gene expression data is reconstructed in 3D, digital space. They support myriad fields of research through interactive visualisation and analysis, providing a valuable source of information for many model organisms including *C. elegans* (Wormbase), frogs (Xenbase), *Drosophila*, and zebrafish."

Comment #2:

In the description of Giotto, it is unclear how spatial information is used "to infer cell type enrichment scores for downstream analyses"

Response:

The sentence has been clarified as follows.

Page 12. Lines 384-386

"A different approach is found in Giotto, a user-friendly workspace that utilises cell type specific gene signatures to infer cell type enrichment scores for downstream analyses with the capability for integration of spatial information."

Comment #3:

For "Computational approaches for resolving spatial gene expression," it would be useful to add a table that summarizes the input and output of each method.

Response:

Columns describing input and output have been added to Table 2. “Computational tools and associated methods for spatial transcriptomics data analysis and visualisation.”

Comment #4:

I suggest English editing, as there are many places where a singular countable noun was used without an article. Also, several sentences are long and difficult to read. Other examples include "development phase" should be "developmental phase"

Response:

Additional editing has been completed.

Comment #5:

This sentence is still not clear: "Where sc-RNA-seq is subject to technical and biological variations, FISH approaches provide a clear visualisation of gene expression, but lack the numerical evaluation of expression provided by sequencing and rely on making comparative evaluations for quantification." I don't think it is fair to say that FISH lacks numerical evaluation because numerical values can be extracted from images. Maybe the authors can change it to "FISH approaches rely on downstream image analysis to extract numerical measurements of gene expression." Meanwhile, scRNA-seq also requires reads filtering, mapping, and quality control to be converted into numerical measurements. Therefore, I do not think the authors should simply say that FISH is less quantitative than scRNA-seq.

Response:

Reviewer #2's suggestion has been incorporated and the text now reads.

Page 5. Lines 137-139:

“FISH approaches rely on downstream image analysis to extract numerical measurements of gene expression. Meanwhile scRNA-seq also requires reads filtering, mapping, and quality control to be converted into numerical measurements.”

Comment #6:

The response to Question 16 is unsatisfactory. The reviewer meant to emphasize that microarray-based spatial transcriptomics assay is different from NGS. The response said that this assay is a type of NGS, contradicting to the reviewer's comment.

Response:

This distinction has been made clearer in the text to avoid confusion.

Page 11. Lines 339-342:

“The ability to generate histological data which precisely complements the tissue section involved in quantitative analysis is unique to Spatial transcriptomics (microarray-based technology) and not possible with NGS-based methods.”

The discussion of challenges with scaling up these technologies has been expanded.

Comment #7:

The response to Question 21 is unsatisfactory. The reviewer meant that the generalization would not be trivial. However, the rewritten sentence still reads that the generalization is easy after the sequencing cost drops.

Response:

Discussion of challenges to scaling in addition to cost has been expanded.

Pages 15-16. Lines 486-494:

“High resolution, imaging-based approaches require high magnification and fine sectioning necessitating long imaging times, which will protract further with scaling. Methods which employ targeted probes carry the inherent limitation of finite available fluorophores, and will be challenging to scale. Underpinning these challenges is that, as more cells and tissues are analysed with higher resolution, more data points are required, demanding higher computational power, and scalable mathematical models for future 3D visualisation and data interpretation. Spatial technologies vary in their readiness for scaled analysis, and this will prove a key determinant of the lifespan and relevancy of the technology.”